# Comparison of Surgeons’ Assessment of the Extent of Vestibular Schwannoma Resection with Immediate Post Operative and Follow-Up Volumetric MRI Analysis

**DOI:** 10.3390/brainsci13101490

**Published:** 2023-10-22

**Authors:** Hossein Mahboubi, William H. Slattery, Mia E. Miller, Gregory P. Lekovic

**Affiliations:** 1House Institute, Los Angeles, CA 90057, USA; 2Cedars-Sinai Hospital, Los Angeles, CA 90048, USA; 3Department of Neurosurgery, David Geffen School of Medicine, University of California Los Angeles, Los Angeles, CA 90024, USA

**Keywords:** vestibular schwannoma, magnetic resonance imaging, residual tumor, gross total resection, near-total resection, subtotal resection

## Abstract

(1) Background: Incomplete excision of vestibular schwannomas (VSs) is sometimes preferable for facial nerve preservation. On the other hand, subtotal resection may be associated with higher tumor recurrence. We evaluated the correlation between intra-operative assessment of residual tumor and early and follow-up imaging. (2) Methods: The charts of all patients undergoing primary surgery for sporadic vestibular schwannoma during the study period were retrospectively reviewed. Data regarding surgeons’ assessments of the extent of resection, and the residual size of the tumor on post-operative day (POD) one and follow-up MRI were extracted. (3) Results: Of 109 vestibular schwannomas meeting inclusion criteria, gross-total resection (GTR) was achieved in eighty-four, near-total (NTR) and sub-total resection (STR) in twenty-two and three patients, respectively. On follow up imaging, volumetric analysis revealed that of twenty-two NTRs, eight were radiographic GTR and nine were radiographic STR (mean volume ratio 11.9%), while five remained NTR (mean volume ratio 1.8%). Of the three STRs, two were radiographic GTR while one remained STR. Therefore, of eighteen patients with available later follow up MRIs, radiographic classification of the degree of resection changed in six. (4) Conclusions: An early MRI (POD#1) establishes a baseline for the residual tumor that may be more accurate than the surgeon’s intraoperative assessment and may provide a beneficial point of comparison for long-term surveillance.

## 1. Introduction

Vestibular schwannoma is a slow growing benign neoplasm of the eighth cranial nerve, commonly referred to as acoustic neuroma. The tumors are rare (1:100,000 incidence) but are the most common tumor of the temporal bone. VS usually presents with hearing loss; less common presenting symptoms include vestibulopathy, facial numbness, headache, or other neurologic signs and symptoms.

Recent decades have been witness to significant changes in the management of vestibular schwannomas. Improvements in understanding the natural history of vestibular schwannomas and enhancements in magnetic resonance imaging (MRI) techniques along with favorable long-term outcomes of radiosurgery have transformed the vestibular schwannoma treatment algorithms [1,2,3,4]. While gross-total resection remains the goal in vestibular schwannoma surgery, incomplete resections have gained popularity where preservation of the integrity of the facial nerve and other structures is otherwise in jeopardy [5].

When incomplete resection is performed, the estimated residual size and shape is used as a baseline for long-term follow-up. The residual tumor is usually observed with serial MRIs for regrowth, which may necessitate treatment by radiosurgery or revision microsurgery. Surgeons’ assessment of a near- or sub-total resection may not be consistent with how much tumor is actually left behind. This intra-operative assessment can be inaccurate as demonstrated in a previous published study [6] in which the authors found no correlation between intra-operative assessment of a near-total resection and post-operative MRI findings. Furthermore, post-surgical changes will occur at the surgical bed over time, which may impede the differentiation of residual tumor from scar tissue. This can become especially problematic if the residual tumor noted on the delayed post-operative MRI is much larger than anticipated. In this situation, it would be challenging to determine whether a small residual tumor underwent significant growth, or a larger residual tumor was left at the initial surgery. 

While practices among vestibular schwannoma surgeons vary, at our institution, an MRI is obtained on post-operative day one in all patients undergoing vestibular schwannoma resection. In this study, we aimed to assess the correlation between the intra-operative assessment of a residual tumor and early post-operative MRI findings by reviewing the data from our cohort of vestibular schwannomas. Furthermore, we compared the immediate and follow-up post-operative images to evaluate for changes in tumor shape and size with time. The results from this study can provide a better understanding of the differences between intra-operative assessment of residual size and post-operative MRI findings. 

## 2. Materials and Methods

This study was approved by our local Institutional Review Board (IRB# SV-018-21). Upon approval, a retrospective chart review was performed on patients who underwent surgery for vestibular schwannomas between October 2017 and October 2019 at our tertiary referral center. Patients with neurofibromatosis type II, revision cases, and those without available pre- or post-operative MRIs were excluded. Data regarding patient demographics, surgical approach, extent of resection (EOR), and intra-operative assessment of residual size were extracted from the charts. Intra-operative degree of resection was reported as gross-total resection (GTR), near-total resection (NTR), or sub-total resection (STR). A GTR was determined when the surgeon was confident of complete tumor removal. NTR was determined when only a thin layer of the tumor capsule was left on the facial nerve or other structures, and this was estimated to be less than 5% of the original tumor volume. Other cases were considered STR. A surgical endoscope was routinely used in cases where the lateral border of the tumor could not be directly visualized (e.g., in the case of the tumor extending laterally to the transverse crest when approaching via the middle fossa).

Patients’ pre- and post-operative MRIs were obtained and analyzed. Our institution’s protocol is to obtain an MRI of the brain with and without contrast, including fat suppression, on post-operative day one. This post-operative MRI was used to evaluate for the presence of any radiographic residual tumor and then compared to the patient’s last available follow up MRI. Horos DICOM image analysis software (v 3.3.5, www.horosproject.org, accessed on 1 October 2019) was used to compute the volume of the pre-operative tumor and those of any residual tumors if present. Volumetric analysis was performed on post-contrast, T1 weighted images. The presence of a residual tumor was confirmed by comparing these images to pre-contrast T1 and fluid-attenuated inversion recovery (FLAIR) as well as other available sequences to rule out hemorrhage or edema. The perimeter method was used to calculate the volumes [7]. The perimeters of tumor or residual tumor were outlined on consecutive images and then the volume of the region of interest (ROI) was calculated using the built-in function from the software. 

Using this volumetric analysis, the degree of resection was defined as radiographic GTR (undetectable residual), radiographic NTR (residual ≤ 5% of the pre-operative tumor volume), or radiographic STR (residual > 5% of pre-operative tumor volume). The correlation between the radiographic and intra-operative assessments of the degree of resection were then compared. The shapes of the residual tumors were further analyzed and classified as linear or nodular. Statistical analysis was performed using SPSS 1.0 (IBM, Armonk, NY, USA). Nonparametric tests were used given the skewed distribution of samples. A *p* value of 0.05 was considered as the threshold of statistical significance. 

## 3. Results

A total of two hundred and three patients undergoing surgery for VSs during the study period were identified, excluding patients with neurofibromatosis type II, revision cases, and those with unavailable MRI images. A total of 109 cases of primary vestibular schwannoma surgeries were analyzed. The mean age was 52.7 years (range 19–79) and 35.8% were males. 

### 3.1. Surgeon’s Asssessment vs. Immediate and Follow Up MRI

#### 3.1.1. Surgical Approach and Surgeon’s Assessment of EOR

The surgical approach was translabyrinthine in sixty-nine (63.3%), middle fossa in thirty (27.5%), retrosigmoid in nine (8.3%), and transotic in one (0.9%). Based on the intra-operative assessments by the senior neurosurgeon, GTR was achieved in eighty-four (77.1%) while a residual was left behind in twenty-five cases (22.9%), consisting of twenty-two NTRs (20.2%) and three STRs (2.8%). The main reason for leaving a residual was severe adherence to the facial nerve in all cases. The intra-operative estimate of residual size varied from 1 to 10 mm.

#### 3.1.2. Immediate Post-Operative MRI Evaluation of EOR

The average pre-operative volume was 3.23 cm^3^ (range 0.02–24.97 cm^3^). The tumors that underwent GTR had a significantly smaller volume (mean 1.93 cm^3^, median 0.55 cm^3^) than those with a residual (mean 7.26 cm^3^, median 4.05 cm^3^, *p* < 0.001). This difference was not statistically significant between tumors that underwent NTR and STR (*p* = 0.9). As detailed in Table 1, MRI confirmed no residual tumor in all cases when the neurosurgeon’s assessment was GTR intra-operatively. Lack of a residual tumor was further confirmed during follow-up MRIs obtained at around 3 and/or 12 months post-operatively.

Volumetric analysis of residual tumors on the immediate post-operative MRI revealed that of twenty-two near-total cases, eight were radiographic GTR and nine were radiographic STR (mean volume ratio 11.9%), while five remained radiographic NTR (mean volume ratio 1.8%). Of the three sub-total cases, two were radiographic GTR while one remained radiographic STR. Of cases with a radiographic residual, seven had linear and eight had nodular residuals. Figure 1 demonstrate examples of these pre- and post-operative tumor images. Overall, radiographic assessment changed the degree of resection in nineteen cases (17.4% of all cases and 76% of cases with a residual).

A sub-analysis was performed on the pre-operative volumes of the tumors for which residual size was underestimated intra-operatively. These tumors had undergone NTR but were recategorized as radiographic STR based on the immediate post-operative MRI. The mean volume of these tumors was 8.93 cm^3^ (median 7.30 cm^3^) compared to a mean volume of 5.27 cm^3^ (median 2.89 cm^3^) in tumors that were considered as radiographic NTRs and GTRs. This difference, however, did not reach statistical significance (*p* = 0.27).

#### 3.1.3. Follow-Up MRI Evaluation of EOR

All patients had follow-up imaging. Follow-up post-operative MRI (ranging 3 to 22 months) was available for eighteen patients with residual tumors (sixteen NTRs and two STRs). Lack of a residual tumor was confirmed via follow-up MRIs obtained around 3 and/or 12 months post-operatively on all patients in whom the intra-operative assessment was consistent with gross-total resection. As seen in Table 2, the resection assessment changed in six cases in between the immediate and delayed MRIs. In two cases, while a residual was seen on the immediate MRI, the delayed MRI was assessed as radiographic GTR. In two cases, while no residuals were seen on the immediate MRI, the delayed MRIs revealed residual tumors (both radiographic NTR; Figure 2). In the other two cases the assessment changed from radiographic STR to NTR.

## 4. Discussion

The present study revealed a disparity between the intra-operative assessment of tumor residual size and post-operative radiographic assessments in some cases. When the surgeon was confident in removing the entire tumor, post-operative MRI confirmed GTR. However, when a residual was left behind, the degree of resection was recategorized in 76% of the cases after the post-operative MRI was reviewed. These findings are consistent with a previous study that evaluated residual tumors on post-operative MRIs in fifty vestibular schwannomas including fifteen NTRs and eight STRs. About 73% of the NTRs were actually noted to be radiographic STR, while all STRs were also radiographic STR [6]. By comparison, the percentage of NTRs being recategorized as radiographic STRs in the present study was lower (nine cases, 41%). In these nine cases where the residual tumor was underestimated, the pre-operative tumor volume was, on average, smaller. Although no statistical significance was reached due to the relatively small sample size, this difference can be clinically significant and suggests that in smaller tumors intra-operative estimation of the residual volume relative to the original tumor may be more difficult.

Surgeons’ intra-operative assessment is mainly subjective and limited by the location of the residual tumor and extent of the surgical field. The definitions of NTR and STR can also vary between institutions. In our practice, a resection is considered near-total if (a) only a thin, linear layer of the tumor capsule is left on the facial nerve (or other structures), and (b) this residual tumor is estimated to be less than 5% of the original tumor volume. While the 5% volume ratio as the threshold to differentiate between NTR and STR has been used in the literature [6,8], other definitions have also been described. One method considers a resection near-total if the residual tumor is less than 25 mm^2^ and 2 mm thick along the facial nerve or brain stem [9,10]. Another method considers residuals less than 5 × 5 × 2 mm as NTR [11]. As seen in these examples, the definitions for NTR are variable and no consensus exists yet.

Our data show that an immediate post-operative MRI can be helpful in establishing the shape and size of the residual and provides a better estimate of the degree of resection than the surgeon’s estimation. In the literature, there is no consensus regarding the timing of the first MRI after vestibular schwannoma resection. A 2005 survey of the American Neurotology Society (ANS) and North American Skull Base Society members revealed that 2.3% of neurotologists and 23.4% of neurosurgeons obtained an MRI on post-operative day one, whereas 21.6% of neurotologists and 61.7% of neurosurgeons obtained the first MRI within a year [12]. A more recent survey of the ANS members revealed similarly variable practices in MRI surveillance. Of the survey responders, the first post-operative MRI was obtained by 18.6% during the immediate post-operative inpatient stay, 23.3% at 3 months, 16.3% at 6 months, and the rest at one year or longer. The completeness of resection was found to affect 73.8% of the responders’ decision about the timing of the first MRI. Among these surgeons, 25.8% indicated that they would obtain the first imaging study earlier than their usual one-year MRI if the excision was incomplete [13].

While the timing and interval of the first post-operative MRI varies between surgeons and institutions, an early MRI during the post-operative inpatient stay could provide additional information to the surgeon. This immediate MRI is obtained on day one after the surgery at our institution. On the MRIs obtained months after the surgery, it may be difficult to differentiate scar tissue from a growing residual tumor [11,14]. An MRI obtained on post-operative day one does not include these nonspecific enhancements. The patient in Figure 2 (#6 in Table 2) had an extremely small residual about 1 mm and the immediate MRI confirmed the lack of a large residual tumor. While the later MRIs showed a larger area of enhancement than anticipated, comparison with the immediate post-operative MRI resulted in more confidence in the degree of resection and differentiation from later non-specific enhancements due to dural inflammation and/or connective tissue formation. As seen in Figure 2, the noted area of enhancement did not change in size from 3 to 15 months post-operatively. Although some studies have shown that these nonspecific enhancements can disappear with time, they may persist in other cases and continue to enhance after 5 years or longer [15,16]. This persistent enhancement can in turn result in a need for prolonged MRI surveillance. The recent survey of the ANS members revealed that a higher percentage of surgeons would follow linear enhancements for longer than 5 years and 14.3% would never stop ordering surveillance MRIs [13]. In this context, an early post-operative MRI could allow surgeons to evaluate the degree of resection and the presence of a residual tumor more accurately, and prior to any scar tissue formation. We believe that most of the changes between POD#1 and three months are attributable to enhancement seen with scarring. Hence, we anticipate that the three-month MRI may actually overestimate the extent of the residual tumor. In addition, the thickness of the obtained images can potentially result in missing small or thin residuals and a radiographic GTR should not obviate the need for continued surveillance in cases with a known residual. Furthermore, the current study was limited by its sample size and future studies will need to follow a larger cohort and compare the long-term appearances of residual tumors to those in the immediate post-operative MRIs.

## 5. Conclusions

Preservation of the facial nerve integrity may necessitate leaving a small residual vestibular schwannoma that is then followed by serial MRIs. Intra-operative assessment of the residual size may be inaccurate given its subjective nature. In our series, post-operative MRI changed the assessment of the degree of resection in 76% of the cases where a residual tumor was left behind. While there is no consensus on the timing of the first post-operative MRI among institutions, an early post-operative MRI establishes a baseline for the residual tumor prior to scar formation and could provide a critical point of comparison for long-term surveillance.

## Figures and Tables

**Figure 1 brainsci-13-01490-f001:**
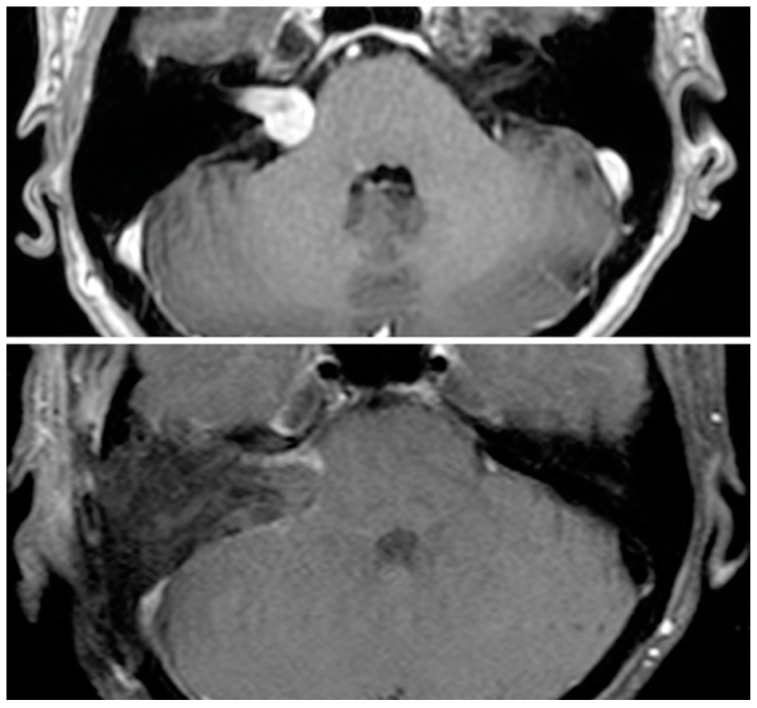
Pre-operative MRI (**top** panel) shows right sided vestibular schwannoma that was resected (**bottom** panel) using a translabyrinthine approach. While this was assessed as a near-total resection, post-operative volumetric analysis classified the residual as subtotal resection (7.9% of original tumor volume).

**Figure 2 brainsci-13-01490-f002:**
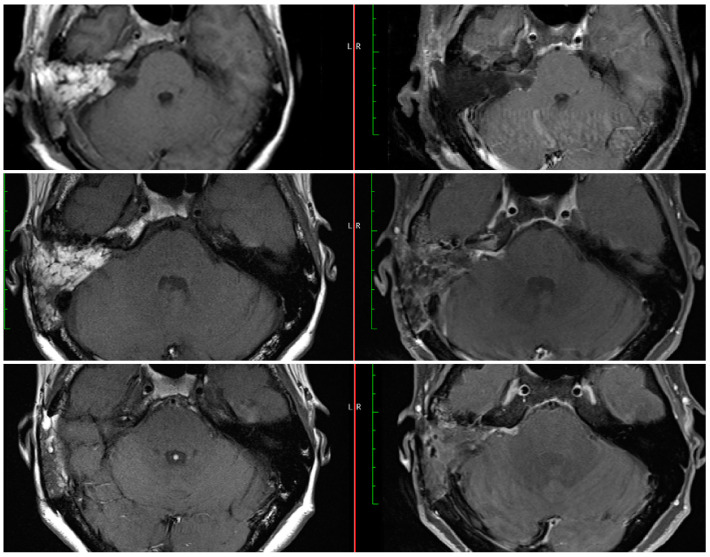
Post-operative MRIs at day 1 (**top**), 3 months (**middle**), and 15 months (**bottom**) after NTR of a right sided vestibular schwannoma. The intra-operative residual estimate was 1 mm. The immediate MRI was a radiographic GTR, but the delayed MRIs were radiographic NTR with stable appearance of the area of enhancement.

**Table 1 brainsci-13-01490-t001:** Surgical resection based on intra-operative versus radiographic assessment.

Intra-Operative Assessment	Immediate Post-Operative MRI			
	Gross-Total	Near-Total	Sub-Total	Total
Gross-total	84 (100%)	0 (0%)	0 (0%)	84
Near-total	8 (36%)	5 (23%)	9 (41%)	22
Sub-total	2 (67%)	0 (0%)	1 (33%)	3
Total	94	5	10	

**Table 2 brainsci-13-01490-t002:** Comparison of the residual volume ratios between immediate and delayed post-operative MRI.

	Age/Sex	Intra-Operative Assessment	Residual Volume Ratio		Change in Radiographic Assessment
			Immediate MRI	Delayed MRI	
1	29 F	NTR	R-NTR	R-NTR	No
2	55 F	NTR	R-STR	R-GTR	Yes
3	35 F	NTR	R-NTR	R-NTR	No
4	53 F	NTR	R-NTR	R-GTR	Yes
5	62 F	NTR	R-NTR	R-NTR	No
6	58 F	NTR	R-GTR	R-NTR	Yes
7	61 F	NTR	R-STR	R-STR	No
8	57 F	NTR	R-GTR	R-GTR	No
9	50 M	NTR	R-GTR	R-GTR	No
10	65 M	NTR	R-GTR	R-NTR	Yes
11	40 F	NTR	R-NTR	R-NTR	No
12	68 F	NTR	R-STR	R-STR	No
13	52 F	NTR	R-STR	R-NTR	Yes
14	25 F	NTR	R-STR	R-NTR	Yes
15	54 F	NTR	R-GTR	R-GTR	No
16	54 F	NTR	R-GTR	R-GTR	No
17	71 M	STR	R-STR	R-STR	No
18	54 M	STR	R-GTR	R-GTR	No

R-GTR: Radiographic GTR.

## Data Availability

The data presented in this study are available on request from the corresponding author. The data are not publicly available due to concern for patient privacy.

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
