# Peer review of "Comparison of Surgeons’ Assessment of the Extent of Vestibular Schwannoma Resection with Immediate Post Operative and Follow-Up Volumetric MRI Analysis"

_brainsci, 2023, doi:10.3390/brainsci13101490_

Round 1

Reviewer 1 Report

You present your data on a very interesting clinical question. Unfortunately despite a large series of over 100 schwannomas you have follow up data only on 18 of them and your results are not very informative nor they can be related with clinical outcomes (i.e., what is the very definition of success in schwannoma surgery?)   Some comments for future improvements: - do you use (angled) endoscopes to check for residual disease at the end of the dissection? - SPSS version 1.0, really? please do report at least the standard deviation of the volumetric measurements and mean or median months for the follow up time - can you provide some data regarding facial nerve function and audiometric postoperatively?  - "The study was conducted according to the guidelines of the 237 Declaration of Helsinki, and approved by the Institutional Review Board (or Ethics Committee) of 238 St. Vincent’s Hospital (protocol code XXX and date of approval)." something is missing here...

Reviewer 2 Report

Abstract:

Between lines 1 and 2 of the background section, the authors could add another sentence to better develop the link between these two sentences.

In the methods section:

Data regarding the surgeon’s assessment extent of resection, and residual size 14 of the tumour on POD#1 and follow-up MRI was extracted. The authors should assume the reader is not familiar with this acronym (POD) and should spell it out. I can see that later in the abstract this has been defined, so they could move it up.

The authors define three categories of resection.

The conclusion is also something that would be reasonably expected.

Introduction:

The authors start the introduction with some fundamental background of the disease. If the authors could also add some stats about the disease this would be useful. It would also be useful to provide more information about this disease in the first paragraph. 

The rationale behind STR as opposed to GTR is also mentioned.

In the following paragraph, the management is discussed and the intra-operative assessment may be inaccurate warranting post-surgical MRI-based assessment. 

An interesting point is made in lines 45-48 in that it is not clear whether a large post-operative growth may be due to significant growth or large residual disease post-resection. The authors could speculate and also use evidence from the literature to attempt to address this issue. Tumour biopsies and IHC and molecular tests would of course help greatly but in the absence of those and only relying on imaging what can be suggested?

So, the authors seek to find a link based on evidence from patient data post-operation.

the results from this study can provide a better understanding of the differences between intra-operative assessment of residual size and post-operative MRI findings. This sentence starts with a lowercase letter.  

Methods:

The authors could provide any institutional review board certificate numbers if relevant. As with many of these medical studies, it would be very useful if the authors could provide a table of the summary characteristics of the patients. This not only helps them get a better sense of the clinical characteristics of these patients, but it will also support powering future meta-analyses.

The general description of the difference between total and near-total resection has been ok.

Patients’ pre- and post-operative MRIs were obtained and analyzed. Our institution’s protocol is to obtain an MRI of the brain with and without contrast, including fat suppression, on postoperative day one. This post-operative MRI was used to evaluate for the presence of any radiographic residual tumour and then compared to the patient’s last available follow-up MRI.  This completely makes sense.

The description of the volumetry method is generally good.

The results section could indeed do with a flowchart that shows the number of patients who were selected and then the rest of the process.

The data in Table 1 is quite useful. It might be beyond the scope of the study but it would be interesting to know if the degree of resection immediately after the operation is linked to any of the clinicopathological characteristics of the patient (e.g., age, gender, stage, etc.).

An aesthetic suggestion about Figure 1 please align the figure subsections and try to get them the same size. Also, the figure legends must appear below the figures.

The lack of a residual tumour was further confirmed on follow-up MRIs obtained at around 3 and/or 12 months post-operatively.  The authors could explain why there is a wide gap between 3 and 12, where some patients may get imaged at 3 months, the others at 12 or both, this could lead to missing tumour growth.

Overall, 119 radiographic assessments changed the degree of resection in 19 cases (17.4% of all cases 120 and 76% of cases with a residual).  What does this suggest?

What determines whether the degree of resection assessed by MRI changes or not? The authors could speculate.

As mentioned some type of flowchart or table would allow us to follow the results much better, since there are 3 groups and from those then other groups branch off.

In the figures, the authors could use asterisks or arrowheads to point to specific features in images, and the description of what those are can be added to the figure legend and if significant in the main text.

In the follow-up MRI section, the authors could explain why only a small fraction of the patients got a follow-up and not the whole cohort or at least a larger fraction of the cohort.

As seen in Table 2, the resection assessment changed in 6 cases in between the immediate and delayed MRIs. The authors could explain that more.

In 2 cases, while a residual was seen on the immediate MRI, the delayed MRI was assessed as radiographic GTR. The authors could explain that the residual disease detected after treatment is only due to the underestimation made by the surgeon or could it also be significant growth. I guess the key point is time. 1 day after and 3 months… the latter is quite a lot of time for a tumour to grow. The authors could try to delineate the difference between the assessment changing right after surgery with 3 months or 12 months after surgery, at present, this is not clear in this study.

It would be visually helpful to use some type of colour coding with Table 2, for example, the binary yes/ no option could get a red/green colour.

A competent discussion and conclusion section, thanks.

Minor issues here and there

Round 2

Reviewer 1 Report

Thank you for making the manuscript clearer.

Reviewer 2 Report

The authors have provided a point-by-point response to my comments, thanks

Minor editing required